# Reservoir Lithology Identification Based on Multicore Ensemble Learning and Multiclassification Algorithm Based on Noise Detection Function

**DOI:** 10.3390/s23041781

**Published:** 2023-02-05

**Authors:** Menglei Li, Chaomo Zhang

**Affiliations:** 1College of Geophysics and Petroleum Resources, Yangtze University, Wuhan 430100, China; 2Key Laboratory of Exploration Technologies for Oil and Gas Resources, Ministry of Education, Yangtze University, Wuhan 430100, China

**Keywords:** noise detection function, multicore ensemble learning multiclassification algorithm, SAMME algorithm, reservoir lithology identification

## Abstract

Reservoir lithology identification is an important part of well logging interpretation. The accuracy of identification affects the subsequent exploration and development work, such as reservoir division and reserve prediction. Correct reservoir lithology identification has important geological significance. In this paper, the wavelet threshold method will be used to preliminarily reduce the noise of the curve, and then the MKBoost-MC model will be used to identify the reservoir lithology. It is found that the prediction accuracy of MKBoost-MC is higher than that of the traditional SVM algorithm, and though the operation of MKBoost-MC takes a long time, the speed of MKBoost-MC reservoir lithology identification is much higher than that of manual processing. The accuracy of MKBoost-MC for reservoir lithology recognition can reach the application standard. For the unbalanced distribution of lithology types, the MKBoost-MC algorithm can be effectively suppressed. Finally, the MKBoost-MC reservoir lithology identification method has good applicability and practicality to the lithology identification problem.

## 1. Introduction

In terms of conventional reservoir lithology identification, the crossplot method [1] is relatively easy to implement and is a manual interpretation method. It can also obtain intuitive results when studying lithology analysis. By using the core data, the chart can be corrected, and then the intersection coordinates of different lithology can be identified on the crossplot, that is, the distribution areas of different lithology can be visually observed, and the boundaries of different lithology can be visually observed. However, because there are not a large number of logging curves that can be used, it is also vulnerable to human factors, it is easy to omit some layers, and it takes human resources and time [2]. The multiple-kernel learning (MKL) method can be used to improve the recognition accuracy as well as to reduce the interferences between the logging curves, thus reducing the errors. Multiple kernel learning (MKL) has received extensive attention in the field of machine learning. It is a promising data mining method. MKL mainly uses the linear combination of multiple kernel functions to solve challenging problems such as heterogeneous or irregular data, uneven distribution of samples, etc. At present, in the field of machine learning, nonlinear classification problems can be effectively solved through kernel techniques [3,4]. However, due to the possible diversity and uncertainty of sample data, it has become an important trend to combine multiple kernel functions to obtain better generalization performance, which has made the multicore learning method receieve widespread attention [5,6,7]. The multicore model is a very flexible learning model based on the kernel method. Recent theoretical studies and practical applications have also shown that the performance of learning models can be improved by using multicore models, and interpretable decision functions can be obtained at the same time. In the multicore learning framework, the problem of data expression in feature space is transformed into the problem of kernel weight selection through kernel function, and different base kernel selection strategies constitute different multicore models [8]. Traditional multicore learning methods usually regard solving the linear combination of multiple kernel functions as an optimization problem, and then focus on the optimization algorithm to achieve better efficiency and accuracy, but the large amount of calculation and difficult convergence are still the main shortcomings of traditional multicore learning methods. Multiple kernel boosting (MKBoost) is AdaBoost applied to multicore learning [9]. AdaBoost is an iterative algorithm. The core idea of AdaBoost is to train different classifiers (weak classifiers) for the same training set, and then combine these weak classifiers to form a stronger final classifier (strong classifier). In order to reduce the loss function during the training process, MKBoost-MC will continuously improve the weight of the wrongly classified samples and reduce the weight of the correctly classified samples. Compared with the pure data set, the samples mixed with noise are more likely to be misclassified. According to the update method of the original weight, the misclassified noise samples will be given higher weight, resulting in overfitting and reducing the classification performance of the classifier [10,11]. Therefore, improving the robustness of MKBoost-MC to noisy samples in data sets has become an important research topic.

## 2. MultiKernel Ensemble Learning Multiclassification Algorithm for Noise Detection Function

### 2.1. MKBoost Algorithm

The MKBoost algorithm uses multiple kernel functions to train weak classifiers, and also adopts the idea of AdaBoost to construct the final combined kernel function, which skillfully avoids solving complex optimization problems. In order to avoid complex optimization problems in multicore learning, Xia [12] proposed two types of multicore integrated learning algorithms, namely deterministic MKBoost and stochastic MKBoost. These two types of algorithms not only ensure the accuracy of the algorithm, but also improve the efficiency of the algorithm and avoid complex optimization problems in multicore learning. The core idea of the deterministic MKBoost algorithm is to ensure that each kernel function will be adopted in each iterative training. MKBoost determines the weight distribution of the kernel function after each training by repeatedly training multiple kernel functions and iterating continuously. After each training, it increases the weight of the wrong classification and reduces the weight of the correct classification, so as to pay attention to the samples that are difficult to classify correctly. Random MKBoost randomly extracts from the set of kernel functions, and the probability of extraction is completely determined by the previous training results. Although compared with the deterministic MKBoost algorithm, it improves the operation efficiency of the algorithm, but also sacrifices the accuracy of the algorithm to a certain extent. Therefore, this paper will focus on the deterministic MKBoost algorithm.

### 2.2. SAMME Algorithm

Initially, AdaBoost was proposed as a binary classification algorithm. With the deepening of research, scholars proposed a variety of AdaBoost multiclassification algorithms, such as SAMME, AdaBoost.Cost, AdaBoost.M1, AdaBoost.MH, AdaBoost.MR etc. The SAMME algorithm proposed by Zhu [13] is also the algorithm used in sklearn when AdaBoost is used for multiclassification problems. The SAMME algorithm is different from AdaBoost. MH and AdaBoost.MR and other AdaBoost multiclassification algorithms; the SAMME algorithm can be directly applied to multiclassification problems without decomposing the multiclassification problem into two classification problems. The SAMME algorithm uses the method of adding items and setting the maximum number of iterations in AdaBoost to achieve multiclassification. Therefore, this paper will introduce the idea of the SAMME algorithm into the MKBoost algorithm to solve the multiclassification problem.

### 2.3. Noise Detection Function

MKBoost, like AdaBoost, is easily disturbed by noisy data and has poor robustness when training noisy data [14]. In order to reduce the loss function during the training process, MKBoost will continuously improve the weight of the wrongly classified samples and reduce the weight of the correctly classified samples. Compared with the pure data set, the samples mixed with noise are more likely to be misclassified. According to the update method of the original weight, the misclassified noise samples will be given higher weight, resulting in overfitting and reducing the classification performance of the classifier. Therefore, it is increasingly important to improve the robustness of MKBoost to noisy samples in data sets. Cao [15] proposed the ND_AdaBoost algorithm to solve the problem of the noise-sensitivity of the AdaBoost algorithm. The ND_AdaBoost algorithm mainly uses the nearest-neighbor algorithm as the noise detection function to identify the mixed noise samples in the data set. The experimental results show that the noise-detection function can effectively identify the noise samples in the data set mixed with noise. If the classification result of the kernel classifier optimized in the round of the MKBoost algorithm is input into the noise detection function, the noise detection function will find the nearest samples around the sample point according to the Euclidean distance, of which *j* = 1, 2, , *K*, and combine them into a nearest-neighbor sample set Kt. The error rate formula for distinguishing sample points is shown as:(1)$$\mu_{t}(Z_{i})=\frac{\sum_{Z_{i}\in K^{t}}I(f_{t}(x_{j})f=y_{j})}{K}$$

In Equation (1), is the prediction result label of the base classifier. The formula for calculating the average noise probability of all sample points is shown in Equation (2):(2)$${\bar{\mu }}_{t}=\frac{1}{N}\sum_{i=1}^{N}\mu_{t}(x_{i},y_{i})$$

The relationship between each point and the average probability is compared. If $\mu_{t}(x_{i},y_{i})>\mu_{t}$, the sample point is a noise sample, and the sample noise label is $\emptyset_{t}(x_{i})=-1$*∅t*(*x_i_*) = −1. The sample points are added to the noise sample set, that is, $N=SN\cup (x_{i},y_{i})$. If $\mu_{t}(x_{i},y_{i})>\mu_{t}$, the sample points are added to the non-noise sample set, that is, $NNS=NNS(x_{i},y_{i})$. ND_AdaBoost effectively solves the problem of AdaBoost noise sensitivity by introducing the KNN algorithm to build a noise detection function. Mkboost is the same idea applied to the AdaBoost algorithm. Inspired by the AdaBoost algorithm, this paper will also introduce the noise detection function to identify the noise samples in the data set and improve the robustness of the algorithm.

## 3. Reservoir Lithology Identification Simulation

### 3.1. Application Principle of Algorithm

The logging process will be affected by many external factors, resulting in the logging data containing interference data. Taking spontaneous potential logging as an example, under normal circumstances, the formation water and the well mud have different degrees of mineralization. Because of the different degrees of mineralization, there is a potential difference. In the logging response, the permeable layer will show as abnormal [16,17]. Therefore, when dividing the sand shale section, the permeability layer can be divided by using the spontaneous potential curve. If the rock stratum has high permeability, then, compared with the mud solution, the formation water has a greater degree of mineralization. At this time, the analysis of spontaneous potential can confirm that it is a negative anomaly. If it has poor permeability, the analysis of spontaneous potential can confirm that it is a small negative anomaly [18]. Therefore, when judging whether it is sandstone, natural potential can be used, which can also clearly reflects the shale content, etc. However, natural potential is easily affected by industrial stray current in the process of logging. When large-scale power equipment works near the well pad, such as DC generators and electric welding machines, it will have a certain impact on the logging results.

### 3.2. Data Denoising

Before using the original logging curve to identify the reservoir lithology, this paper first uses the wavelet threshold method to denoise the logging data and improve the signal-to-noise ratio in the data. The threshold selection adopts the global unified threshold, as shown in Equation (3):(3)$$\lambda =\sigma^{\sqrt{21nN}}$$

In Equation (3), σ is the intermediate values of the absolute values of the first level wavelet decomposition coefficients, 0.6745 is the adjustment coefficient of the standard variance of Gaussian noise, and N is the length of the signal.

### 3.3. Normalization Processing

When calculating, because the logging data have different dimensions, it needs to be normalized so that it can be between [0, 1]. By constructing a unified standard, normalization can be achieved, as shown in Equation (4):(4)$$X^{*}=\frac{X-X_{\min}}{X_{\max}-X_{\min}}$$

In Equation (4), *X*^*^ is the normalized logging data, which is the range of original logging data [0, 1]. The result *X*_min_ is the minimum value of the original logging parameters, and *X*_max_ is the maximum value of the original logging parameters. General logging curves can be normalized by Equation (4). If the curve has nonlinear characteristics, such as reservoir permeability and resistivity, Equation (5) needs to be used to realize normalization:(5)$$X=\frac{\mathrm{lg}X^{*}-\mathrm{lg}X_{\min}^{*}}{\mathrm{lg}X_{\max}^{*}-\mathrm{lg}X_{\min}^{*}}$$

### 3.4. Kernel Function Selection

For the kernel function, in order to realize the mapping of data in high-dimensional space, so that the classification space can be formed, it is necessary to select the kernel function. The selection of the kernel function requires the analysis of training data, and the selection of multiple kernel functions, so that the kernel function can have a high classification effect. In this paper, the maximum value of parameter C is 100, the minimum value is 1, and the maximum number of iterations is set to 500. Each kernel function reservoir lithology is identified ten times. The prediction result takes the average value of the accuracy of reservoir lithology identification ten times. The prediction result of the radial basis function kernel is: medium conglomerate, 62%; pebbly fine sandstone, 63%; coarse sandstone, 68%; gravelly medium sandstone, 71%; medium sandstone, 73%; fine conglomerate, 77%; mudstone, 79%; silty mudstone, 71%. The prediction results of polynomial kernel function are: medium conglomerate, 57%; fine sandstone, 64%; pebbly fine sandstone, 66%; mudstone, 68%; silty mudstone, 68%; pebbly medium sandstone, 70%; coarse sandstone, 72%; medium sandstone, 76%. Through experiments, it can be seen that the radial basis function has advantages in the identification of reservoir lithology of medium conglomerate, fine conglomerate, pebbly medium sandstone, mudstone, and silty mudstone, and the polynomial kernel function has certain advantages in the identification of coarse sandstone, medium sandstone, and pebbly fine sandstone. Therefore, aiming at the problem of reservoir lithology identification of the multicore integrated learning model, the kernel function mainly selects the radial basis function and the polynomial kernel function.

### 3.5. Experimental Results and Analysis of Reservoir Lithology Identification

In order to verify that MKBoost-MC can effectively identify the type of reservoir lithology, the logging data are taken as the input variable, and then the traditional SVM is used for training and comparative study of reservoir lithology identification, and the results are selected according to the kernel function. The penalty coefficient C in SVM is set to 10, the maximum number of iterations T is 100, and K in the noise detection function is 8. Taking the effective logging data of the whole well section of wells A and B as the training data and the effective logging data of well C as the test data, 20 recognition experiments are carried out for each algorithm, and the average value of the recognition accuracy is taken to verify the effectiveness of the algorithm for reservoir lithology recognition. The experimental results are shown in Table 1 and Table 2.

It can be seen from the table that the number of wrong samples predicted by MKBoost-MC is 127, the absolute value of error is 10.21%, and the time required is 15 minutes. The absolute value of error in the prediction results is low, but the prediction takes more time. The accuracy of MKBoost-MC in lithology prediction is higher than that of the traditional SVM algorithm. The main reason is that although the data required for this experiment have been denoised by the wavelet threshold, and the denoised data have met the requirements of manually distinguishing lithology by the crossplot method and other methods, this part of the logging data are still mixed with some noise data that cannot be removed. For the noise-sensitive multicore integrated learning multiclassification algorithm, it will affect the robustnessof the algorithm and reduce the accuracy of reservoir lithology recognition [19]. When there is a certain proportion of noise data in the logging curve, MKBoost-MC can effectively improve the generalization of the MKBoost algorithm, which is better than the SVM algorithm. It proves that the weight update can be well adapted to the problem of reservoir lithology identification using logging curves, and MKBoost-MC can more effectively reduce the impact of noise. In terms of prediction time, compared with the SVM algorithm, due to the influence of algorithm iteration, MKBoost-MC takes a long time to calculate, while the SVM algorithm takes less time. From the perspective of time efficiency, MKBoost-MC has no advantage. It should be pointed out that in the work of reservoir lithology identification, the speed of MKBoost-MC reservoir lithology identification is much higher than that of manual processing. Comprehensively comparing the accuracy and time efficiency of reservoir lithology identification, the weakness in time efficiency of MKBoost-MC can be appropriately ignored. From the perspective of engineering application, for the logging parameters with a small amount of noise data, the accuracy of MKBoost-MC for reservoir lithology recognition can reach the application standard. For the unbalanced distribution of lithology types, the MKBoost-MC algorithm can be effectively suppressed.

## 4. Conclusions

In this paper, MKBoost-MC is selected to study the logging curve, so as to realize the purpose of lithology prediction of multiple logging curves with noise. In view of the sensitivity of the multicore ensemble learning multiclassification algorithm to noise, a noise detection function is constructed to calculate the probability that the sample points are noise according to each round of training results, and compared with the average probability to determine whether it is noise data. Then, the idea of the SAMME algorithm is introduced to realize the purpose of transforming binary classification into a multiclassification algorithm. Finally, a new weight-update method is proposed, and the MKBoost-MC model is used to reduce the impact of noise data on the next iteration. The simulation results of reservoir lithology recognition based on MKBoost-MC show that the robustness of the algorithm is effectively improved based on MKBoost-MC. By establishing a single-core classifier to recognize the reservoir lithology, according to the reservoir lithology recognition results based on MKBoost-MC, the kernel function with the best effect is selected, and then a noise-resistant MKBoost-MC reservoir lithology recognition model is established. It is proposed to use the wavelet threshold to preliminarily process the original logging data, and then input the processed data set into the model for training. The effectiveness of the MKBoost-MC reservoir lithology identification model in the field of reservoir lithology identification is further verified by comparative tests, which further proves that the MKBoost-MC reservoir lithology identification model has certain practical significance.

Although the research results show that this paper has achieved the expected goal in dealing with a small amount of noise mixed in logging data, and the proposed multicore integrated learning and multiclassification algorithm of the noise detection function can also be well applied, there are still two problems to be solved. Firstly, although the robustness of the algorithm to noise is effectively improved by introducing the noise detection function, at the same time, it also further affects the training efficiency of the algorithm, and the training speed needs to be further improved. Secondly, the algorithm needs to use wavelet threshold and other noise reduction methods for preliminary noise reduction in reservoir lithology identification, which increases the complexity of the whole identification process. In future research, it is hoped that the algorithm can directly identify the original collected data and avoid using other noise reduction algorithms.

## Figures and Tables

**Table 1 sensors-23-01781-t001:** Lithology prediction results of SVM algorithm.

Type	Medium Conglomerate	Fine Conglomerate	Coarse Sandstone	Medium Sandstone	Pebbly Fine Sandstone	Sandstone Sandstone	Mudstone	Silty Mudstone
Medium conglomerate	191	-	-	-	17	3	-	4
Fine conglomerate	-	245	5	-	-	-	6	-
Coarse sandstone	11	-	179	-	-	2	-	-
Medium sandstone	-	16	-	123	-	-	-	10
Pebbly fine sandstone	-	-	-	-	130	-	17	-
Pebbly sandstone	-	-	13	9	-	75	-	3
mudstone	5	-	-	-	-	-	274	-
Silty mudstone	-	-	-	-	6	-	-	297

**Table 2 sensors-23-01781-t002:** Lithology prediction results of MKBoost-MC algorithm.

Type	Medium Conglomerate	Fine Conglomerate	Coarse Sandstone	Medium Sandstone	Pebbly Fine Sandstone	Sandstone Sandstone	Mudstone	Silty Mudstone
Medium conglomerate	191	-	-	-	17	3	-	4
Fine conglomerate	-	245	5	-	-	-	6	-
Coarse sandstone	11	-	179	-	-	2	-	-
Medium sandstone	-	16	-	123	-	-	-	10
Pebbly fine sandstone	-	-	-	-	130	-	17	-
Pebbly sandstone	-	-	13	9	-	75	-	3
mudstone	5	-	-	-	-	-	274	-
Silty mudstone	-	-	-	-	6	-	-	297

## Data Availability

No new data were created or analyzed in this study. Data sharing is not applicable to this article.

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
