# Peer review of "Reservoir Lithology Identification Based on Multicore Ensemble Learning and Multiclassification Algorithm Based on Noise Detection Function"

_sensors, 2023, doi:10.3390/s23041781_

Round 1
Reviewer 1 Report
This paper addresses interesting question of lithology identification, it needs to address following points
1. Abstract must reflect the outcomes of the paper
2. Add in line 13, Introduction. In this section add part describing importance of current research and comparison study with another researchers.
3. Add nomenclature or definition of all variables in all equations.
4. Section 3 presents short description of the algorithms without any link to the main target of lithology identification in examined cases.
5. Table 1 depicts results of SVM and table 2 for KM algorithms. I can not understand 100% similarity of results????
6. This algorithms results need core data validation. This is very serious matter.
7. Explain why using wells A, B for training and well C for test, I believe the wells have different lithologies.
8. Conclusion must be shorten and limited only to research outcomes.
Author Response
1.Abstract must reflect the outcomes of the paper
Finally the MKBoost-MC reservoir lithology identification method has good applicability and 11 practicality to the lithology identification problem.
2.Add in line 13, Introduction. In this section add part describing importance of current research and comparison study with another researchers.
Added description of the research importance of the method
- Section 3 presents short description of the algorithms without any link to the main target of lithology identification in examined cases.
The response characteristics of the logging curves are different due to different lithologies, so this method can effectively avoid the interaction between the logging curves
- Explain why using wells A, B for training and well C for test, I believe the wells have different lithologies.
The lithology of wells A, B and C is basically the same, so wells A and B are used for training, and the test done by well C
- Conclusion must be shorten and limited only to research outcomes.
Briefly modified for the conclusion

Reviewer 2 Report
Dear editor,
I have finished my review of Li and Zhang manuscript entitled Reservoir lithology identification based on multi-core ensemble learning and multi classification algorithm based on noise detection function, submitted to the Sensors journal.
Firstly I would like to emphasize that the manuscript is clear to read and well-written. Concerning the submitted manuscript, I would only like to point out that several important references were left aside. This mainly occurs in the introduction - I will point out some examples below.
However, my major point, which made it very difficult for me to evaluate the manuscript, is that the dataset was not presented or even not superficially discussed. Please, if possible, answer these questions: It is not clear the data set used (How is and from where is your dataset?); it is not clear the logs utilized (what logs you used? GR? NPHI? RHOB?).
This is the main dataset information - Lines 146 - 149: Taking the effective logging data of the whole well section of well A and B as the training data and the effective logging data of well C as the test data, 20 recognition experiments are carried out for each algorithm, and the average value of the recognition accuracy is taken to verify the effectiveness of the algorithm for reservoir lithology recognition. The experimental results are shown in Table 1 and Table 2.
My main knowledge is on geology, and this manuscript focuses on geological reservoirs without any information addressing this issue. So for me, I could not understand how the authors reached their results and what is their data source.
I believe that the authors have in their hands a very good manuscript. However, I am afraid about how they reached their results. Following these points, I would like to recommend major reviews. I would appreciate in revise a revised view of this manuscript.
References: (Introduction)
(Lines 15-16) It can also obtain intuitive results when studying lithology analysis.
(Lines 31-33) Recent theoretical studies and practical applications have also shown that the performance of learning models can be improved by using multi-core models, and interpretable decision functions can be obtained at the same time. (please cite the references that provided this information).
With best regards,
Author Response
It is not clear the logs utilized (what logs you used? GR? NPHI? RHOB?).
The logging curves involved in this experiment are mainly using SP, GR, NPHI, RHOB, RT curves.

Round 2
Reviewer 1 Report
It is Ok now
Reviewer 2 Report
Dear editor,
The manuscript is now suitable for publication.
Best regards.